# Effects of Long-Term Use of Organic Fertilizer with Different Dosages on Soil Improvement, Nitrogen Transformation, Tea Yield and Quality in Acidified Tea Plantations

**DOI:** 10.3390/plants12010122

**Published:** 2022-12-26

**Authors:** Jianghua Ye, Yuhua Wang, Jiaqian Kang, Yiling Chen, Lei Hong, Mingzhe Li, Yun Jia, Yuchao Wang, Xiaoli Jia, Zeyan Wu, Haibin Wang

**Affiliations:** 1College of Tea and Food, Wuyi University, Wuyishan 354300, China; 2College of Life Science, Fujian Agriculture and Forestry University, Fuzhou 350002, China; 3College of Life Science, Longyan University, Longyan 364012, China

**Keywords:** tea tree (*Camellia sinensis*), soil acidification, sheep manure, nitrogen transformation, tea yield and quality

## Abstract

In this study, sheep manure fertilizers with different dosages were used for five consecutive years to treat acidified tea plantation soils, and the effects of sheep manure fertilizer on soil pH value, nitrogen transformation, and tea yield and quality were analyzed. The results showed that soil pH value showed an increasing trend after a continuous use of sheep manure fertilizer from 2018 to 2022. After the use of low dosage of sheep manure fertilizer (6 t/hm^2^–15 t/hm^2^), tea yield, the content of tea quality indicators (tea polyphenols, theanine, amino acid, and caffeine) and soil ammonium nitrogen content, ammoniating bacteria number, ammoniating intensity, urease activity and protease activity showed increasing trends and were significantly and positively correlated to soil pH value, while the related indexes showed increasing and then decreasing trends after the use of high dosage of sheep manure fertilizer (18 t/hm^2^). Secondly, the nitrate nitrogen content, nitrifying bacteria number, nitrifying intensity, nitrate reductase activity, and nitrite reductase activity showed decreasing trends after the use of low dosage of sheep manure fertilizer and showed significant negative correlations with soil pH value, while the related indexes showed decreasing trends after the use of high dosage of sheep manure and then increased. The results of principal component and interaction analysis showed that the effects of sheep manure fertilizers with different dosages on tea yield and quality were mainly based on the transformation ability of ammonium nitrogen and nitrate nitrogen in the soil, and the strong transformation ability of ammonium nitrogen and the high ammonium nitrogen content in the soil were conducive to the improvement of tea yield and quality, and vice versa. The results of topsis comprehensive evaluation and analysis showed that the most influential effect on the fertilization effect was the ammonium nitrogen content in the soil and long-term treatment with 15 t/hm^2^ of sheep manure fertilizer had the highest proximity to the best fertilization effect. This study provided an important practical basis for the remediation and fertilizer management in acidified tea plantation soils.

## 1. Introduction

Tieguanyin (*Camellia sinensis*) tea tree is native to Anxi County, Fujian Province, China, and is a perennial evergreen plant. Tea tree is an acidophilic plant, and its suitable soil pH value for growth is from 4.5 to 5.5. An either too acid or too alkaline environment will have a negative impact on the growth of tea tree [1,2]. As a cash crop, tea trees are harvested with mainly shoots and young leaves and therefore have a high demand for fertilizers during growing processes [3]. In recent years, due to the increase in economic benefits of tea, tea farmers have used a large amount of chemical fertilizers in order to obtain higher economic benefits, and the annual use of chemical fertilizers exceeded about twice the recommended use in tea plantations. The large amount of chemical fertilizers did not effectively guarantee the yield and quality of tea, but rather exacerbated soil acidification [4,5]. In 2018, 363 tea plantation soils in nine major tea-producing townships in Anxi County were collected and found that 37.67% of the tea plantation soils were acidified (pH < 4.5) and unsuitable for planting tea trees [6]. In 2021, 5285 tea plantation soils from 23 townships throughout Anxi County were collected. It was found that the proportion of tea plantation soils with pH < 4.0 was 28.38%, the proportion of soil pH between 4.0 and 4.5 was 40.06%, and the proportion of acidified soil in the overall tea plantation soils reached 68.44% [7]. It could be seen that soil acidification of tea plantations in Anxi County was extremely serious, and serious acidification damaged the soil texture of tea plantations, reduced the yield and quality of tea trees and limited the development of tea industry [8,9].

Organic fertilizers are mainly derived from plants or animals, which not only provide comprehensive nutrients for crops, but also have a long-term fertilization effect, increase and renew soil organic matter, promote microbial reproduction, improve soil physicochemical properties and biological activity and are an important guarantee for ecological cultivation [10,11,12,13]. The use of organic fertilizers, instead of chemical fertilizers, is of great significance for improving soil acidificaton, maintaining soil health and achieving organic ecological agriculture. Animal manure can be made into environmentally and friendly organic fertilizers through fermentation and decomposition, which can not only provide nutrients for plant growth effectively, but also improve soil texture [14].

Therefore, many scholars have also conducted a large number of studies on effects of animal organic fertilizers on the improvement of soil acidification and properties, such as sheep manure organic fertilizer [15,16], cow manure organic fertilizer [17], and pig manure organic fertilizer [18,19]. All these studies have shown that animal manure fertilizers can effectively improve soil, promote crop growth and improve the yield and quality of crops. In terms of organic fertilizer use in tea plantation, our research team, in the early stage, used organic fertilizer such as sheep manure fertilizer, cow manure fertilizer, and pig manure fertilizer to treat tea plantation soil and found that sheep manure fertilizer was the most effective for the improvement of tea yield and quality [20] and, based on this, proceeded to conduct a follow-up study. Another major reason is that, at present, the adoption of organic fertilizers instead of chemical fertilizers is also being promoted in Chinese tea plantations to ensure that soil ecology of tea plantation can be suitable and healthy, and the choice of fertilizers is mainly made by using well-rotted sheep manure. However, the preliminary study by our team confirmed that the use of sheep manure fertilizer to improve soil acidification in tea plantations had some effects, but how to control the dosage used, especially how to choose the dosage for long-term application in order to achieve better results? And what the mechanism of sheep manure fertilizer improvement on acidified soil is and its effect on tea yield and quality formation. So far, few studies have been reported in these aspects. Accordingly, this study selected tea plantations with severely acidified soil as the research object and treated with different dosages of sheep manure fertilizer to analyze changes in soil pH value, as well as nitrogen transformation and their effects on tea yield and quality for five consecutive years (2018–2022), with a view to providing practical guidance for the improvement of acidified tea plantation soil and the regulation of tea plantation fertilization.

## 2. Results and Discussion

### 2.1. Effects of Sheep Manure Fertilizer on Soil pH and Tea Yield

Tea trees are acidophilic crops. Soil acidification is highly likely to affect the normal growth of tea trees, especially when the soil pH is less than 4.5, and the growth of tea trees will be severely hindered [6]. It has been reported that when pH < 4.5, the root growth tissue of tea tree was broken, the number and area of new root systems were significantly reduced, and root growth was inhibited, which in turn led to a significant decrease in plant height and biomass of tea tree [21]. Secondly, the damage to the root system of tea trees directly affects the nutrient uptake and utilization efficiency of tea trees, which in turn reduces the yield and quality of tea trees [9,22]. The use of organic fertilizers to improve soil acidification can effectively reduce the degree of soil acidification and ensure crop yield [15]. For example, the use of rape cake organic fertilizer to improve acidified soils can effectively improve soil acidification and crop yield and quality [23]. A similar effect can be achieved by using cow manure organic fertilizers to improve acidified soils [17]. In this study, we used sheep manure fertilizer as a material to analyze the improvement effects of different dosages of sheep manure on soil acidification in tea plantation for five consecutive years and its effect on tea yield. The results showed (Figure 1) that sheep manure fertilizer could effectively improve soil acidification in tea plantation, and the soil pH of tea plantation changed significantly with the increase of sheep manure use (6 t/hm^2^–18 t/hm^2^) and the extension of use time (2018−2020), especially when the amount of sheep manure fertilizer reached 18 t/hm^2^ and the pH value of tea plantation soil reached 5.96 in 2022 (Figure 1A). The results of tea yield analysis showed (Figure 1B) that the use of sheep manure fertilizer was conducive to the tea yield, but the excessive use of sheep manure fertilizer reduced tea yield. Taking the tea yield in 2020 as an example. The trend of different dosages of sheep manure fertilizer on tea yield showed 15 t/hm^2^ (5634 kg/hm^2^) > 12 t/hm^2^ (4726 kg/hm^2^) > 18 t/hm^2^ (4425 kg/hm^2^) > 9 t/hm^2^ (4108 kg/hm^2^) > 6 t/hm^2^ (3725 kg/hm^2^). The results of the simulated curve analysis of soil pH value and tea yield showed (Figure 1C) that soil pH value was positively correlated with the tea yield from 2018 to 2022 after the use of low-dosage sheep manure fertilizer (6 t/hm^2^–15 t/hm^2^). In contrast, tea yield showed a trend of increasing and then decreasing after the use of high-dosage sheep manure fertilizer (18 t/hm^2^). It could be seen that sheep manure fertilizer improved the acidification of tea plantation soil, but the range of soil pH value suitable for tea tree growth was between 4.5 and 5.5, and the most suitable pH value was between 5.0 and 5.5 [1]. After the excessive use of sheep manure fertilizer, soil acidification was effectively improved and the pH value increased in the early stage, but with the extension of time used, the soil pH value exceeded the suitable range of tea trees growth, which affected the normal growth of tea trees and reduced the tea yield.

### 2.2. Effect of Sheep Manure Fertilizer on Tea Quality

The economic benefit of tea tree is usually evaluated comprehensively by the yield and quality of tea. Yield is the basis of economic benefit, quality is the enhancement of economic benefit, and improving tea quality on the basis of guaranteeing the yield is the embodiment of the maximizing the economic benefit of tea [24,25]. Tea quality was usually evaluated by the content of inclusions in tea leaves [26]. Tea polyphenols, theanine, and caffeine were important components that constituted the flavor and aroma characteristics of tea leaves, while amino acids play an important role in forming the umami and sweet taste of tea soup [27,28,29]. Many scholars usually evaluated tea quality by measuring the contents of tea polyphenols, theanine, caffeine, and amino acids in tea leaves, with high content resulting in high quality and vice versa [30,31,32]. In this study, the effects of different dosages of sheep manure fertilizer and soil pH value on tea quality were evaluated using the above indexes, and the results showed (Figure 2 and Figure 3) that the use of sheep manure fertilizer was beneficial to improve tea quality, and when the dosage of sheep manure fertilizer was used at 6 t/hm^2^–15 t/hm^2^, the tea quality indexes (tea polyphenols, theanine, amino acids, and caffeine) showed an increasing trend, and when the dosage of sheep manure was 18 t/hm^2^, the content of tea quality indexes showed an increasing trend followed by a decreasing trend. For example, when the dosage of sheep manure fertilizer was used at 15 t/hm^2^, the tea polyphenol content increased from 223.15 mg/g to 281.26 mg/g with the increase of the useful year. When the dosage of sheep manure fertilizer was used at 18 t/hm^2^, the content of tea polyphenols increased from 239.28 mg/g to 252.38 mg/g and then decreased to 226.17 mg/g. Secondly, in 2022, the trend of the effects of different dosages of sheep manure fertilizer on tea polyphenol content showed 15 t/hm^2^ (281.26 mg/g) > 12 t/hm^2^ (251.37 mg/g) > 9 t/hm^2^ (238.16 mg/g) > 18 t/hm^2^ (226.17 mg/g) > 6 t/hm^2^ (185.24 mg/g). Furthermore, the results of the simulated curve analysis of soil pH value and tea quality showed that (Figure 3) soil pH value was positively correlated with tea quality after continuous use of low dosages of sheep manure fertilizer (6 t/hm^2^–15 t/hm^2^), while with the increase of soil pH value tea quality showed an increasing and then decreasing trend after the use of high dosage of sheep manure (18 t/hm^2^). It could be seen that sheep manure fertilizer improved the soil acidification of tea plantation, but long-term excessive use will negatively affect the growth of tea trees and the quality of tea leaves. Therefore, the use of sheep manure fertilizer should be controlled.

### 2.3. Effects of Sheep Manure Fertilizer on Ammonium and Nitrate Nitrogen Contents in Soil

Tea trees are mainly harvested for young buds and leaves; therefore, their growth has a high demand for fertilizer, especially nitrogen, and the formation of tea quality is closely related to soil nitrogen content [3]. Soil nitrogen pool is mainly composed of organic and inorganic nitrogen, while the nitrogen absorbed by plants is mainly inorganic nitrogen in the soil, including ammonium (NH_4_^+^-N) and nitrate (NO_3_^−^-N) [33]. Tea trees were ammonia-loving plants, and ammonium nitrogen was beneficial for the root development of tea tree and the accumulation of free amino acids in tea leaves, which facilitated the growth and quality of tea tree [34]. However, the available nitrogen content in the soil was significantly and positively correlated with the yield and quality of tea, especially the ammonium nitrogen content [35]. This study found that the content of ammonium nitrogen in tea plantation soil increased significantly after the use of different dosages of sheep manure fertilizer, while the opposite was true for nitrate nitrogen content. Further analysis revealed that the content of ammonium nitrogen in tea plantation soil increased significantly and the content of nitrate nitrogen decreased significantly after a continuous use of low dosage sheep manure fertilizer (6 t/hm^2^–15 t/hm^2^) (Figure 4A). Taking 2022 as an example. When the dosages of sheep manure were 6 t/hm^2^, 9 t/hm^2^, 12 t/hm^2^, and 15 t/hm^2^, soil ammonium nitrogen content was the highest compared with in other years, which were 4.89 mg/kg, 5.99 mg/kg, 6.72 mg/kg, and 11.87 mg/kg, respectively, while soil nitrate content was the lowest compared with those in other years, which were 45.89 mg/kg, 39.87 mg/kg, 34.21 mg/kg, and 27.04 mg/kg, respectively. After a continuous use of high dosage of sheep manure fertilizer (18 t/hm^2^), the soil ammonium content increased first and then decreased, while the nitrate content change was opposite (Figure 4A).

The transformation of soil nitrogen form was closely related to soil pH value. When soil pH value decreased, soil nitrification increased and nitrate nitrogen content increased, and ammonium nitrogen content decreased [36,37]. This study further simulated the relationship between soil pH value and the content of ammonium and nitrate nitrogen after the use of different dosages of sheep manure fertilizer from 2018 to 2022. The results showed (Figure 4B,C) that after a continuous use of low dosages of sheep manure fertilizer (6 t/hm^2^–15 t/hm^2^), the soil ammonium nitrogen content tended to increase and nitrate nitrogen content tended to decrease with the increase of soil pH value, while after a continuous use of high dosage of sheep manure fertilizer (18 t/hm^2^), ammonium nitrogen content increased first and then decreased with the increase of soil pH value, and nitrate nitrogen content change was opposite. It could be seen that sheep manure fertilizer improved soil acidification, increased the ammonium nitrogen content and decreased nitrate nitrogen content, which was conducive to the growth of tea trees, but excessive use was not conducive to the growth of tea trees.

### 2.4. Effects of Sheep Manure Fertilizer on the Number and the Intensity of Soil Nitrogen Transforming Microorganisms

Nitrogen transformation and its contents of different forms in soil are closely related to the number and intensity of microorganisms related to nitrogen transformation [38,39]. For example, ammonifying bacteria facilitate the conversion of organic nitrogen and mineralize nitrogen into ammonium for plant absorption and utilization [40,41], while nitrifying bacteria convert ammonium nitrogen into nitrate nitrogen in soil, which could be absorbed and utilized by plants on the one hand, but easily led to nitrogen loss in soil on the other hand [42,43]. In this study, we found that with the increase of useful years, the number of soil ammonifying bacteria showed a significant increase and the number of nitrifying bacteria showed a significant decrease after the use of low-dosage sheep manure fertilizer (6 t/hm^2^–15 t/hm^2^). However, the number of soil ammonifying bacteria increased first and then decreased after the use of high-dosage sheep manure fertilizer (18 t/hm^2^), while the opposite was true for nitrifying bacteria (Figure 5A). The results of biochemical intensity analysis showed that the intensity of soil ammoniation was consistent with the change trend of the number of ammonifying bacteria after the use of different dosages of sheep manure, while the nitrification intensity was consistent with the change trend of the number of nitrification bacteria in the soil (Figure 6A). It could be seen that low dosage of sheep manure fertilizer was beneficial to improve soil acidification and increase the number of ammonifying bacteria in the soil, which in turn increased the soil ammoniation intensity and ammonium nitrogen content. Although high dosage of sheep manure fertilizer was beneficial to improve soil acidification, long-term excessive use was detrimental to the accumulation of soil ammonium nitrogen.

Soil pH value is closely related to the number of soil microorganisms, and changes in pH value significantly affect the number of microorganisms related to nitrogen transformation and their intensity in soil, which in turn affects the nitrogen transformation capacity of soil [44]. This study further simulated and analyzed the relationship between the soil pH value and the number and the intensity of microorganisms after the use of different dosages of sheep manure from 2018 to 2022. The results showed that after a continuous use of low dosage of sheep manure fertilizer (6 t/hm^2^–15 t/hm^2^), the number and the intensity of ammonifying bacteria in soil showed an increasing trend with the increase of soil pH value (Figure 5B,C), while the number and the intensity of nitrifying bacteria showed a decreasing trend (Figure 6B,C). After a continuous use of high dosage of sheep manure fertilizer (18 t/hm^2^), the number and the intensity of ammonifying bacteria in soil showed an increasing trend followed by a decreasing trend with the increase of soil pH (Figure 5B,C), while the number and the intensity of nitrifying bacteria showed a decreasing and then increasing trend (Figure 6B,C). Nitrifying bacteria were reported to survive more easily and be reproduced at a faster rate in acidic soils, while soil acidification is highly likely to inhibit the reproduction of ammonifying bacteria [45,46]. Although there are some differences in the adaptation of different microorganisms to soil pH, the survival of microorganisms is favored within an appropriate pH range, while beyond this range the reproduction of microorganisms is inhibited [40,47]. It could be seen that sheep manure fertilizer improve soil acidification and promote the propagation of soil ammoniating microorganisms, as evidenced by the increase in their number and ammonification intensity, but the long-term excessive use would still negatively affect the propagation of soil microorganisms.

### 2.5. Effects of Sheep Manure Fertilizer on Soil Enzyme Activities

Soil enzyme activity is an important index for evaluating soil quality, reflecting the direction and intensity of biochemical processes and nutrient cycling in the soil [48]. Soil urease is a highly specific enzyme that promotes the hydrolysis of urea to produce ammonia, which is converted to ammonium for plant uptake and utilization [49]. Soil proteases mainly hydrolyze organic nitrogen to amino acids, which are one of nitrogen sources for higher plants [50]. Nitrate reductase and nitrite reductase can convert nitrogen into nitrate [51,52]. This study found that after the use of low dosages of sheep manure fertilizer (6 t/hm^2^–15 t/hm^2^) from 2018 to 2022, it still showed a significant increase in soil urease and protease activities and a significant decrease in nitrate reductase and nitrite reductase activities. However, after high dosage of sheep manure (18 t/hm^2^), soil urease and protease activities first increased and then decreased, while nitrate reductase and nitrite reductase activities were opposite (Figure 7). Further analysis showed that after continuous use of low dosage sheep manure fertilizer (6 t/hm^2^–15 t/hm^2^) from 2018 to 2022, the activities of urease and protease in soil showed an increasing trend with the increase of soil pH value, while the activities of nitrate reductase and nitrite reductase showed a decreasing trend. However, after a continuous use of high dosage of sheep manure (18 t/hm^2^), the activities of soil urease and protease increased and then decreased with the increase of soil pH value, while the activities of nitrate reductase and nitrite reductase tended to decrease (Figure 8). It could be seen that low pH values were favorable to the transformation and accumulation of nitrate nitrogen in the soil, while being unfavorable to ammonium nitrogen.

### 2.6. Principal Component Analysis and Interaction Network Analysis

The results of principal component analysis showed (Figure 9) that the soil and tea indexes measured after the use of different dosages of sheep manure from 2018 to 2022 could be divided into two principal components, with the contribution rate of principal component 1 being the largest, which varied from 84.7% to 97.6%. Further analysis showed that the treatment with low dosage of sheep manure fertilizer (6 t/hm^2^–12 t/hm^2^) mainly concentrated in the negative end of principal component 1, and the treatment with high dosage of sheep manure (15 t/hm^2^–18 t/hm^2^) was mainly concentrated in the positive end of principal component 1. At the same time, the analysis revealed that the indexes concentrated in the negative side of principal component 1 were nitrate nitrogen, nitrifying bacteria, nitrifying intensity, nitrate reductase and nitrite reductase, which were mainly related to soil nitrate transformation. Secondly, interaction network analysis revealed that there was a positive correlation between these indicators (Figure 10). The indexes concentrated at the positive end of principal component 1 were pH value, tea yield, tea polyphenols, theanine, amino acid, caffeine, ammonium nitrogen, ammonifying bacteria, ammoniation intensity, urease, and protease, which were mainly divided into four categories, including soil pH value, tea yield, tea quality, and soil ammonium transformation. Further results of interaction network analysis showed that there was a positive correlation between the above indexes (Figure 10). In addition, it was also found that there were negative correlations between the indexes concentrated at the positive end of principal component 1 and those at the negative end of principal component 1 (Figure 10). It could be seen that the use of high dosages of sheep manure had the best effect on soil acidification improvement, the effect of the use of sheep manure fertilizer with different dosages on tea yield and quality mainly lied in the transformation ability of ammonium nitrogen and nitrate nitrogen in the soil, and the strong transformation ability of soil ammonium nitrogen and the high content of ammonium nitrogen were conducive to the increase of tea yield and quality, and vice versa.

### 2.7. Optimal Effect Analysis of the Topsis Method with Different Fertilization Dosages

Topsis method is a common and effective method in multi-objective decision analysis, also known as the superior−inferior solution distance method, which can be used for multi-dimensional and multi-index comprehensive evaluation [53]. The greater the weight of the index in its algorithm, the stronger the influence of the index on the fertilization effect; the smaller the distance between the positive ideal value and the comprehensive value is, the larger the distance between the negative ideal value and the comprehensive value is, the closer it is to the best fertilization effect. The relative proximity is the degree of proximity between the fertilization effect and the best fertilization after comprehensive evaluation of all indexes, and the greater the relative proximity indicates it is closer to the best fertilization effect.

Therefore, this study used the topsis model to analyze the effect of different indexes on fertilization effects, and the results showed that the greatest effect on the fertilization effect was from the ammonium nitrogen content in the soil, which occupied the greatest weight and of which the content determined tea yield and quality (Figure 11A). Evaluated after 5 years of continuous use of sheep manure fertilizer, the distance between the positive ideal value and the comprehensive value was the smallest at 18 t/hm^2^ (0.052) and was the second at 15 t/hm^2^ (0.055) (Figure 11B); the distance between the negative ideal value and the comprehensive value was the largest at 15 t/hm^2^ (0.081) and was the second at 18 t/hm^2^ (0.062) (Figure 11C). The results of relative proximity analysis showed that the treatment with 15 t/hm^2^ of sheep manure fertilizer for a long time had the highest proximity to the optimal fertilization effect (Figure 11D). It could be seen that suitable dosages of sheep manure fertilizer effectively improved soil acidification in tea plantation and increased the yield and quality of tea.

## 3. Materials and Methods

### 3.1. Test Tea Plantation and Sample Collection

Soil pH values between 4.5 and 5.5 were suitable for planting tea trees, while the soil pH value below 4.0 was severely acidified, which was not suitable for planting tea trees [1,2,8]. Based on our team’s previous research [6,9], the present study was conducted in Tieguanyin tea plantation in Longjuan Township, Anxi County, Quanzhou City, Fujian Province, China (longitude: 117°93’ east; and latitude: 24°97’ north) as the experimental site (Figure 12). The average altitude of the experimental site is 600 m, the average annual rainfall is 1800 mm, the average annual relative humidity is 80%, and the average annual temperature is 18 °C.

The total area of the experimental tea plantation was 5.49 hm^2^, and the tea tree variety was Tieguanyin. The tea tree was 3 years old, and the average pH value of the soil in the tea plantation was 3.38, which was a severely acidified soil, and other indicators are shown in Appendix A. The experimental tea plantation was divided into five areas, namely B1, B2, B3, B4, and B5, areas of which were 0.64 hm^2^, 0.83 hm^2^, 0.85 hm^2^, 1.61 hm^2^, and 0.87 hm^2^, respectively (Figure 12). The tea plantation was harvested only once a year in May each year and fertilized with sheep manure fertilizer in September each year. At the time of tea harvesting in May each year, the tea yield, quality index content and soil pH value, ammonium nitrogen and nitrate nitrogen contents, the number and the intensity of ammonifying bacteria and nitrifying bacteria, soil enzyme activity, etc. were measured to analyze the effects of different dosages of sheep manure fertilizer on them.

### 3.2. Fertilization Treatment Method

The decomposed sheep manure was used as a fertilizer (1.64% total nitrogen, 0.91% total phosphorus, and 0.89% total potassium) to fertilize tea trees in tea plantation, and sheep manure fertilizer purchased from China Inner Mongolia Guwoheni Agricultural Technology Co., Ltd. The experiment was set up in five different areas B1, B2, B3, B4, and B5 of the same tea plantation, and the amount of sheep manure fertilizer used was 6 t/hm^2^, 9 t/hm^2^, 12 t/hm^2^, 15 t/hm^2^, and 18 t/hm^2^, respectively. During the experiment period from 2017 to 2022, only sheep manure fertilizer was used in the tea plantation, and the fertilizer was applied once a year, each time between 1 September 2017 and 5 September 2017 with the first application on 2 September 2017. In May of the following year (tea-picking time), tea yield, quality index content of tea, soil pH value, ammonium nitrogen and nitrate nitrogen contents, the number and intensity of ammoniating bacteria and nitrifying bacteria, and soil enzyme activity were measured for the first time, and thus for 5 consecutive years (2018–2022). Sheep manure fertilizer was used in tea plantation by burying it in trenches on both sides of tea tree planting field (Appendix A).

### 3.3. Sample Collection

Tea leaves are harvested in May each year. Therefore, the yield of tea leaves in B1–B5 tea plantations was measured in May of each year (May 2018, May 2019, May 2020, May 2021, and May 2022) for five consecutive years after different fertilization treatments from September 2017 to September 2022. At the same time, leaves (functional leaves and the second leaf) of tea trees in different areas of tea plantations were also collected and the root circle soil of tea tree. Among them, the collected tea leaves were used for the determination of tea polyphenols, theanine, amino acids, and caffeine contents, and the tea root circle soil was used for the determination of soil pH value, ammonium nitrogen and nitrate nitrogen contents, the number and the intensity of ammonifying bacteria and nitrifying bacteria, and soil enzyme activities. When collecting soil from the root circle of the tea tree, first removed the fallen leaves from the surface of the soil, then digged away the upper layer of soil up to the root of the tea tree and collected the soil around the root of tea tree.

### 3.4. Determination of Soil pH Value

Soil pH value was measured using a pH meter (PB-10; Sartorius, Göttingen, Germany) with a soil and water ratio of 1:2.5, and 5 replicates were used for each sample.

### 3.5. Determination of Tea Yield

Tea yield was determined according to the method of Wang et al. [26]. Specifically, the measurement was carried out in May each year, and the harvesting standard was 3–4 leaves in the middle and small open surface of the standing buds, with an area of 10 m^2^ (1 row, 10 m in length × 1 m in width) for each sample, and all 10 m^2^ were planted with tea trees. The tea yield of each fertilizer dosage was measured in triplicate samples. Based on the measured tea yield, the tea yield per hectare was converted.

### 3.6. Determination of Tea Quality Index

The contents of tea polyphenols, theanine, amino acids, and caffeine were determined in fresh leaves (functional leaves, i.e., second leaves) collected from tea trees with 5 replicates for each sample. Tea polyphenols was extracted and assayed according to the National Standards of the People’s Republic of China–GBT8313-2018: Determination of total polyphenols and catechins content in tea [54]. Theanine was determined according to the National Standard of the People’s Republic of China–GBT23193-2017: Determination of theanine in tea-Using high performance liquid chromatography [55]. Amino acids was determined according to the National standard of the People’s Republic of China (GBT 8314–2013: Tea determination of free amino acids content) using spectrophotometry and reaction with ninhydrin [56]. Caffeine was determined according to the National Standard of the People’s Republic of China-GBT8312-2013: Tea determination of caffeine content [57].

### 3.7. Determination of Ammonium and Nitrate Nitrogen in Soil

The contents of nitrate nitrogen and ammonium nitrogen in soil were determined by ultraviolet spectrophotometry [58]. Briefly, the soil was extracted with a 2 mol/L KCl solution for 1 h and filtered through a 0.45 μm membrane. The ammonium and nitrate contents in the solution were determined by ultraviolet spectrophotometry using a continuous flow analyzer (SA5000, Skalar, Breda, The Netherlands) and converted to soil nitrogen content.

### 3.8. Determination of the Number of Soil Nitrogen Transforming Bacteria

The number of soil nitrogen transforming bacteria was mainly determined as ammonifying bacteria and nitrifying bacteria [59]. Among them, soil ammonifying bacteria were cultured using a liquid peptone ammonification medium with the composition of 0.5 g KH_2_PO_4_, 0.5 g MgSO_4_∙7H_2_O, 5 g peptone, 1 L distilled water, and pH 7.0–7.2; the determination of ammonifying bacteria was performed by the MPN dilution method. Soil nitrifying bacteria were cultured using modified Stephenson’s medium with the composition of 2 g (NH_4_)_2_SO_4_, 0.75 g K_2_HPO_4_, 0.25 g NaH_2_PO_4_, 0.03 g MgSO_4_∙7H_2_O, 5 g CaCO_3_, 0.01 g MnSO_4_∙4H_2_O, 20 g agar, 1 L distilled water, and pH 7.2; the determination of nitrifying bacteria was performed by the plate colony-counting method.

### 3.9. Determination of the Intensity of Soil Biochemical Processes

The intensity of soil biochemical processes mainly determines the intensity of ammonification and nitrification of soil [59]. Soil ammonification intensity was determined by using ammonification bacterial medium culture and the soil suspension inoculation method. Soil ammonification intensity was evaluated by the content of NH_4_-N in 100 mL culture solution. Soil nitrification intensity, using liquid nitrifying bacteria medium culture, was determined by the soil suspension inoculation method. Nitrification intensity = (nitrite content in the original medium-nitrite content in the medium after incubation)/nitrite content in the original medium × 100%. Nitrification intensity was calculated as the disappearance rate of NO_2_-N.

### 3.10. Determination of Soil Enzyme Activity

In this study, four enzymes involved in soil nitrogen transformation, namely urease, protease, nitrate reductase, and nitrite reductase, were determined using an enzyme-linked immunosorbent assay kit. One gram fresh soil was extracted with PBS buffer solution (The soil-to-water volume ratio was 1:10), after that, the Elisa enzyme-linked immunoassay kit (Beijing Huadeboyi biological technology co., LTD., Beijing, China) was used to extract, and the OD value under 450 nm of the extracting solution was detected by a multifunctional enzyme mark (BioTek Synergy2 Gene 5, American). The results of enzyme activity were expressed as the molar mass (μmol) of the enzyme produced per unit volume (L^−1^) and per unit time (min^−1^). The kit principle is to determine the level of enzyme activity in soil samples using a double antibody clamping method. to determine the level of enzyme activity in the soil samples. As an example of nitrite reductase determination, the purified nitrite reductase antibody was coated with microporous plates to produce solid phase antibodies; the extracted test solution was added to the micropores coated with a monoclonal antibody; further, the horseradish peroxidase (HRP)-labeled nitrite reductase antibody was added to form an antibody−antigen−enzyme-labeled antibody complex. Afterwards, the complex was thoroughly washed and added tetramethyl benzidine (TMB) for color development. TMB was converted to blue under the catalysis of HRP enzyme and finally to yellow under the action of acid. The color depth was positively correlated with the nitrite reductase in the sample. Other enzymes were determined with a similar method.

### 3.11. Statistical Analysis

The average value and variance of data were calculated using Excel 2017 software, regression analysis was performed using DPS software v19.05, Heml 1.0 software was used to make heat maps, Origin 2018 software was used to make trend charts, violin charts, and box charts, Rstudio 3.3 software was used to make principal component maps, and Cytoscape_v3.9.1 software was used to make the correlation analysis chart. The entropy weight topsis statistical analysis was performed on the SPSSAU online platform (https://spssau.com/ (accessed on 10 August 2022)) [53].

## 4. Conclusions

This study analyzed the improvement effect of soil acidification in tea plantation with different dosages of sheep manure fertilizer and their effects on tea yield, quality, and soil nitrogen transformation after a continuous use of sheep manure fertilizer from 2018 to 2022. The results showed (Figure 13) that the long-term use of sheep manure fertilizer was beneficial to improve soil acidification and increase soil pH value, which in turn increased the number of soil ammonifying bacteria, ammonification intensity, urease activity, and protease activity, decreased the number of soil nitrifying bacteria, nitrifying intensity, and nitrate reductase activity and finally increased the ammonium nitrogen content in soil, decreased the nitrate nitrogen content and improved the yield and quality of tea. However, the dosage of sheep manure fertilizer needed to be controlled, with low dosages contributing to the overall improvement effect and high dosages doing the opposite. In this study, it was found that the treatment with 15 t/hm^2^ sheep manure fertilizer could effectively improve the acidification of tea plantation soil and increase the yield and quality of tea leaves, which was closest to the optimal fertilization effect. This study provided an important practical basis for soil remediation and fertilizer management in acidified tea plantations.

## Figures and Tables

**Figure 1 plants-12-00122-f001:**
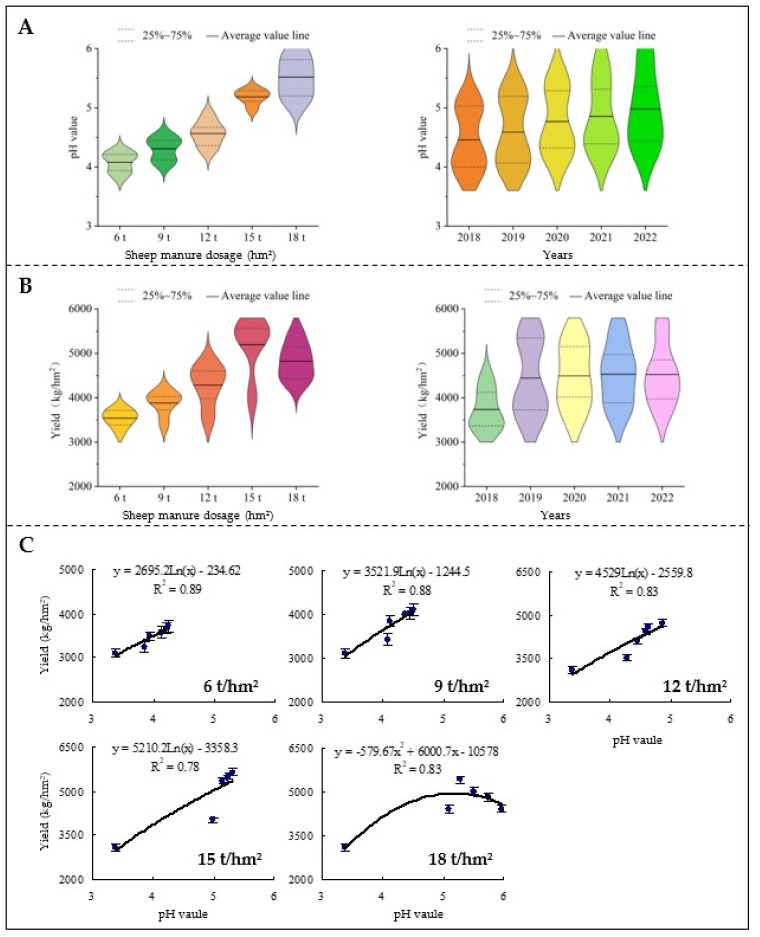
Effects of sheep manure fertilizer on soil pH and tea yield. (**A**) Effects of sheep manure dosage and useful year on soil pH value; (**B**) effects of sheep manure dosage and useful year on tea yield; (**C**) simulation trend analysis of the relationship between soil pH value and tea yield during 5-year application of different dosages of sheep manure fertilizer.

**Figure 2 plants-12-00122-f002:**
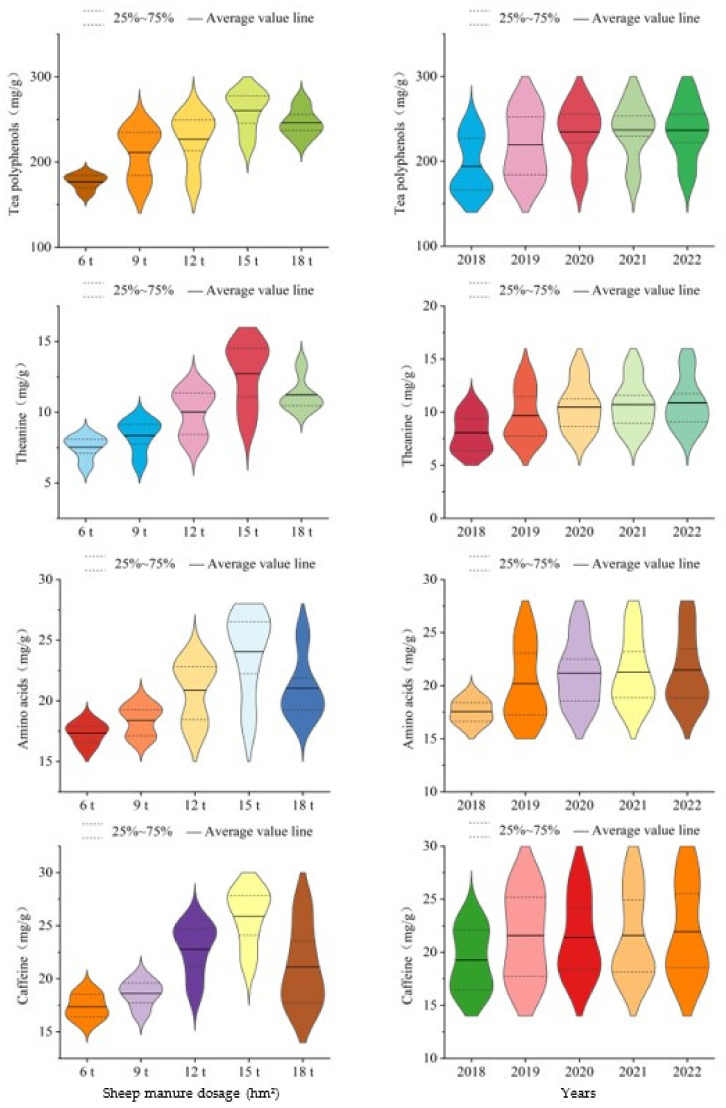
Effects of sheep manure fertilizer on tea quality indexes with the increase of dosage or useful year.

**Figure 3 plants-12-00122-f003:**
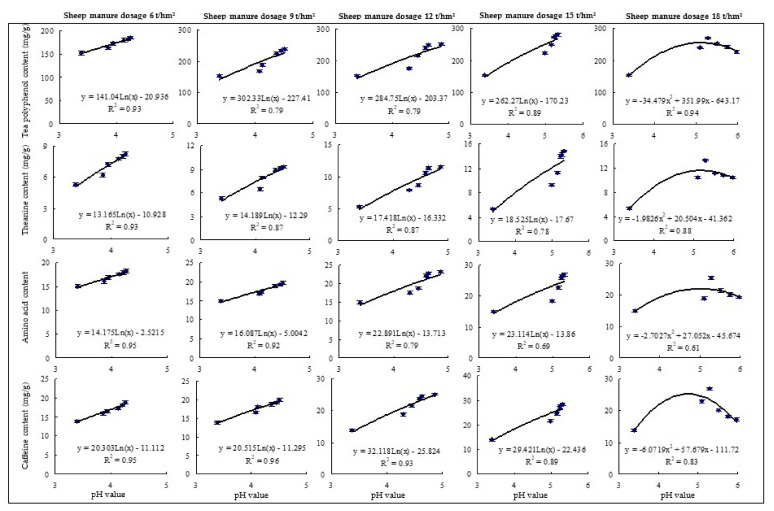
Effects of pH value on tea quality indexes during continuous use of sheep manure fertilizer with different dosages.

**Figure 4 plants-12-00122-f004:**
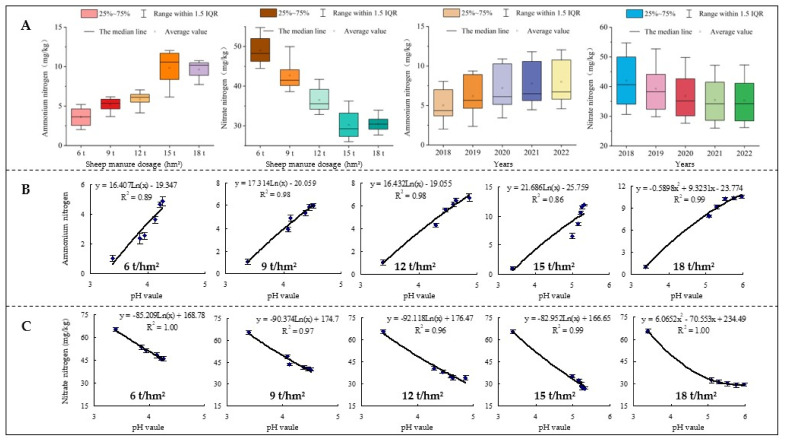
Effects of sheep manure fertilizer and pH value on soil ammonium and nitrate contents in tea plantation. (**A**) Effects of sheep manure fertilizer on soil ammonium and nitrate contents in tea plantation; (**B**) effect of pH value on soil ammonium nitrogen content in tea plantation after a continuous use of sheep manure fertilizer for 5 years; (**C**) effect of pH value on soil nitrate nitrogen content in tea plantation after a continuous use of sheep manure for 5 years.

**Figure 5 plants-12-00122-f005:**
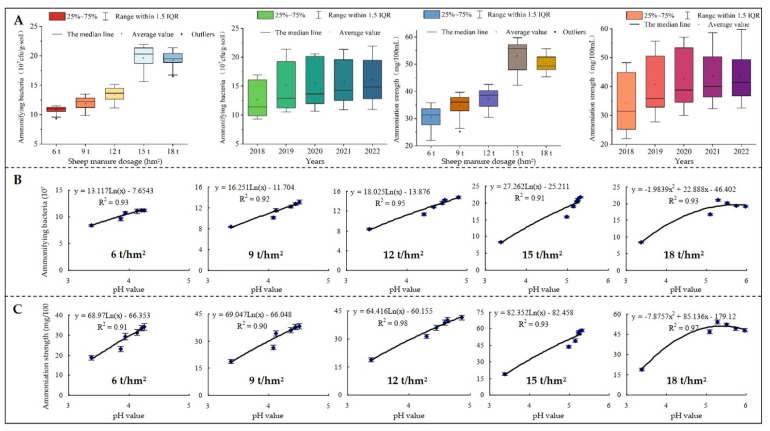
Effects of sheep manure fertilizer and pH value on the number of ammonifying bacteria and ammoniation strength in tea plantation soil. (**A**) Effects of sheep manure fertilizer on the number of ammonifying bacteria and ammoniation strength in tea plantation soil; (**B**) effect of pH value on the number of soil ammonifying bacteria in tea plantation after a continuous use of sheep manure fertilizer for 5 years; (**C**) effect of pH value on soil ammoniation strength in tea plantation after a continuous use of sheep manure fertilizer for 5 years.

**Figure 6 plants-12-00122-f006:**
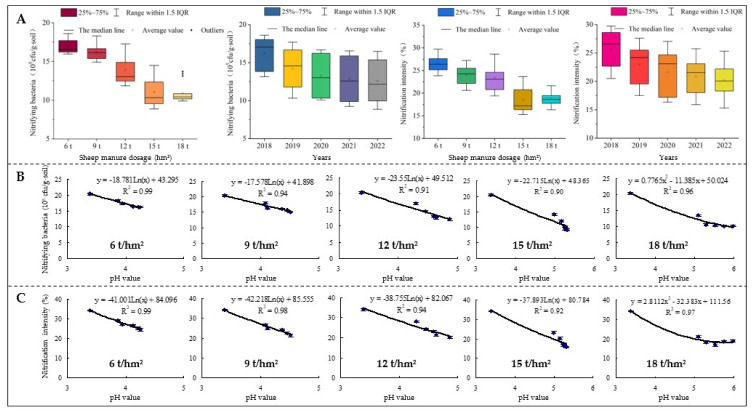
Effects of sheep manure fertilizer and pH value on the number of nitrifying bacteria and nitrification strength in tea plantation soil. (**A**) Effects of sheep manure fertilizer on the number of nitrifying bacteria and nitrification strength in tea plantation soil; (**B**) effect of pH value on the number of soil nitrifying bacteria in tea plantation after a continuous use of sheep manure fertilizer for 5 years; (**C**) effect of pH value on soil nitrification strength in tea plantation after a continuous use of sheep manure fertilizer for 5 years.

**Figure 7 plants-12-00122-f007:**
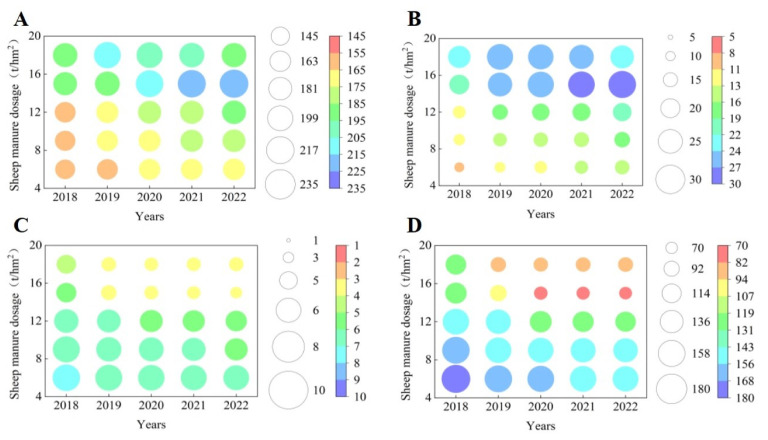
Effects of sheep manure fertilizer on soil enzyme activities in tea plantation. (**A**) Urease (μmol/min/L); (**B**) protease (μmol/min/L); (**C**) nitrate reductase (μmol/min/L); (**D**) nitrite reductase (μmol/min/L).

**Figure 8 plants-12-00122-f008:**
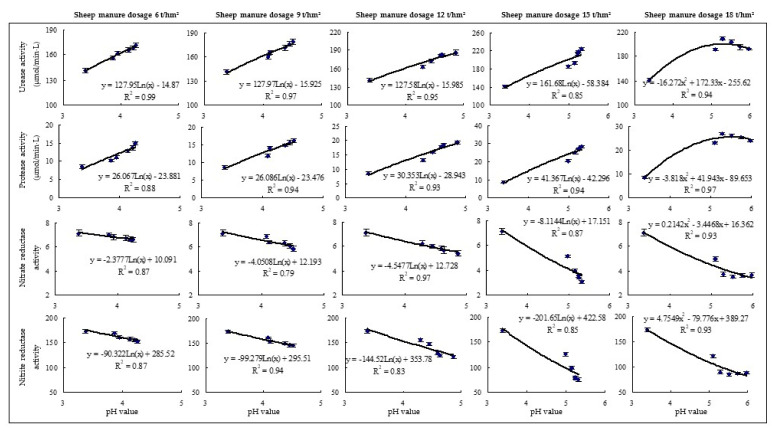
Effects of pH value on soil enzyme activity in tea plantation after a continuous use of sheep manure fertilizer with different dosages.

**Figure 9 plants-12-00122-f009:**
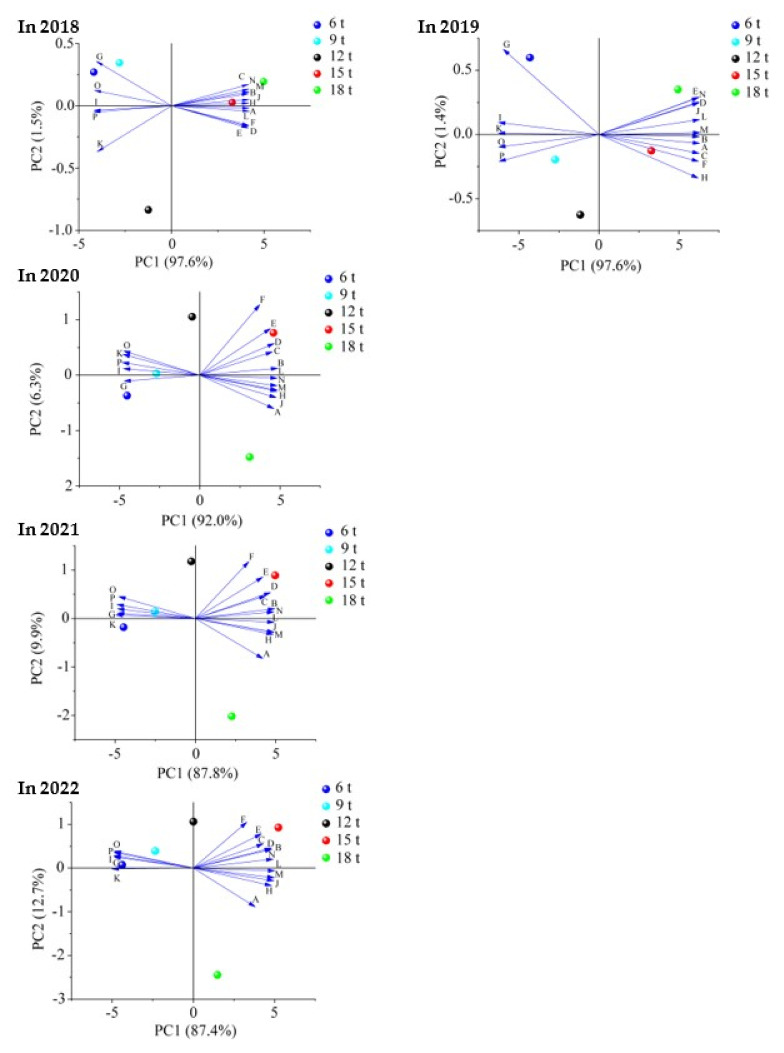
Principal component analysis of soil and tea indexes of different years after the use of sheep manure fertilizer with different dosages. (**A**) pH value; (**B**) tea yield; (**C**) tea polyphenols; (**D**) theanine; (**E**) amino acid; (**F**) caffeine; (**G**) nitrate nitrogen; (**H**) ammonium nitrogen; (**I**) nitrifying bacteria; (**J**) ammonifying bacteria; (**K**) nitrification intensity; (**L**) ammoniation intensity; (**N**) urease; (**M**) protease; (**O**) nitrate reductase; (**P**) nitrite reductase.

**Figure 10 plants-12-00122-f010:**
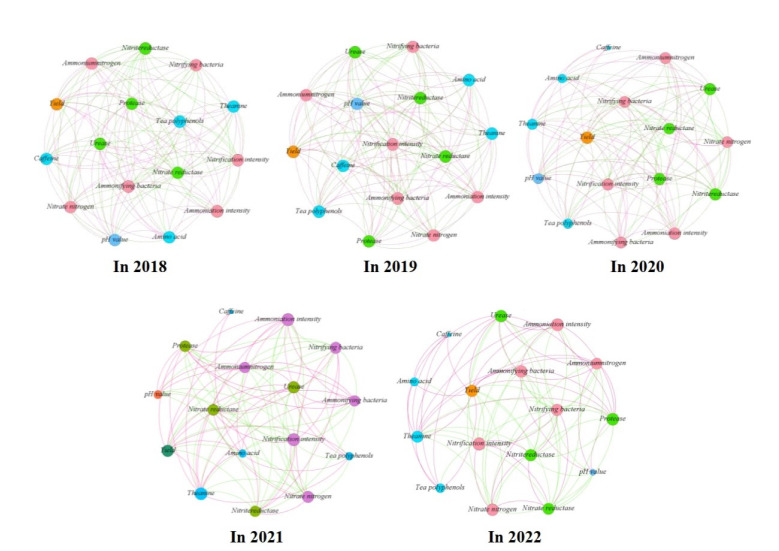
Interaction networks based on correlation analysis of soil and tea indexes of different years after the use of sheep manure fertilizer with different dosages. Note: 

 positive correlation; 

 negative correlation.

**Figure 11 plants-12-00122-f011:**
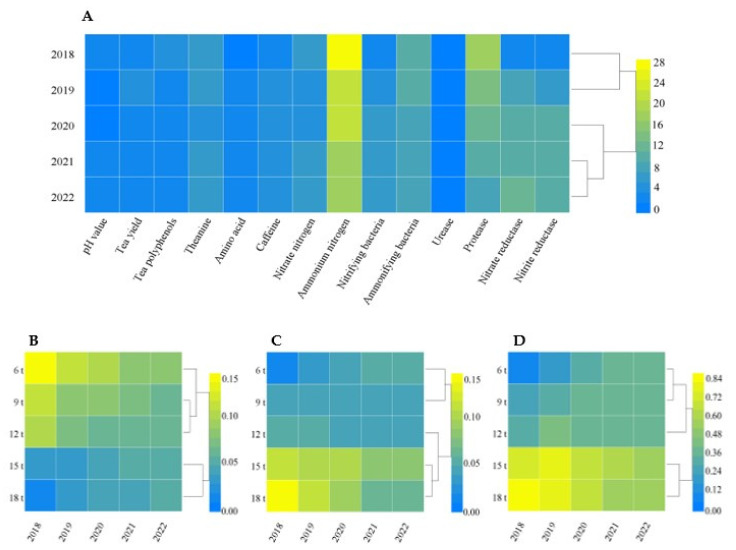
Topsis analysis of comprehensive evaluation of soil and tea tree indexes after long-term use of sheep manure with different dosages. (**A**) Weight of soil and tea tree indicators under different dosages of sheep manure treatment from 2018 to 2022; (**B**) distance synthesis values of positive ideal solutions (D^+^) for each index under different dosages of sheep manure treatment from 2018 to 2022; (**C**) distance synthesis values of negative ideal solutions (D^−^) for each index under different dosages of sheep manure treatment from 2018 to 2022; (**D**) relative proximity between the fertilization effect and the optimal scheme after different sheep manure treatments in 2018–2022.

**Figure 12 plants-12-00122-f012:**
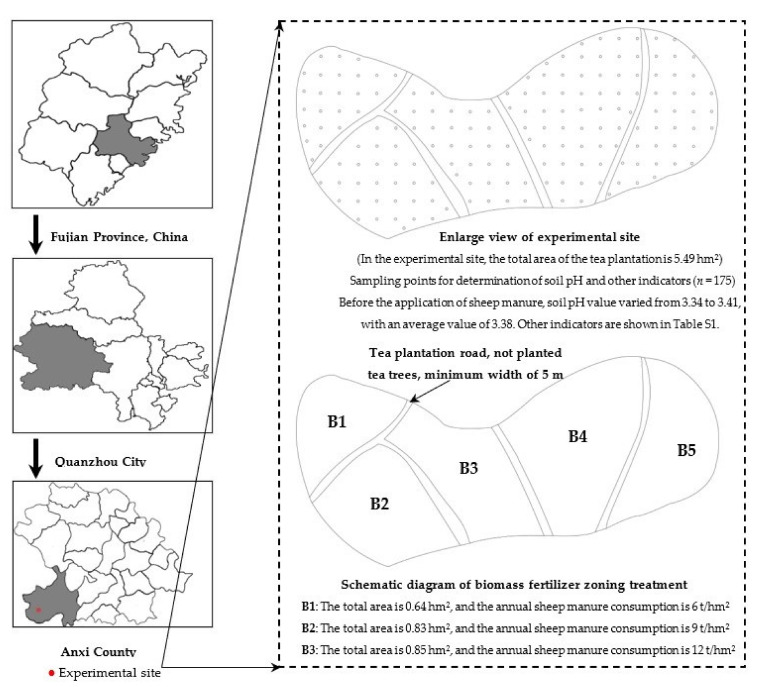
Schematic diagram of the experimental site and tea plantation subdivision from 2017 to 2022.

**Figure 13 plants-12-00122-f013:**
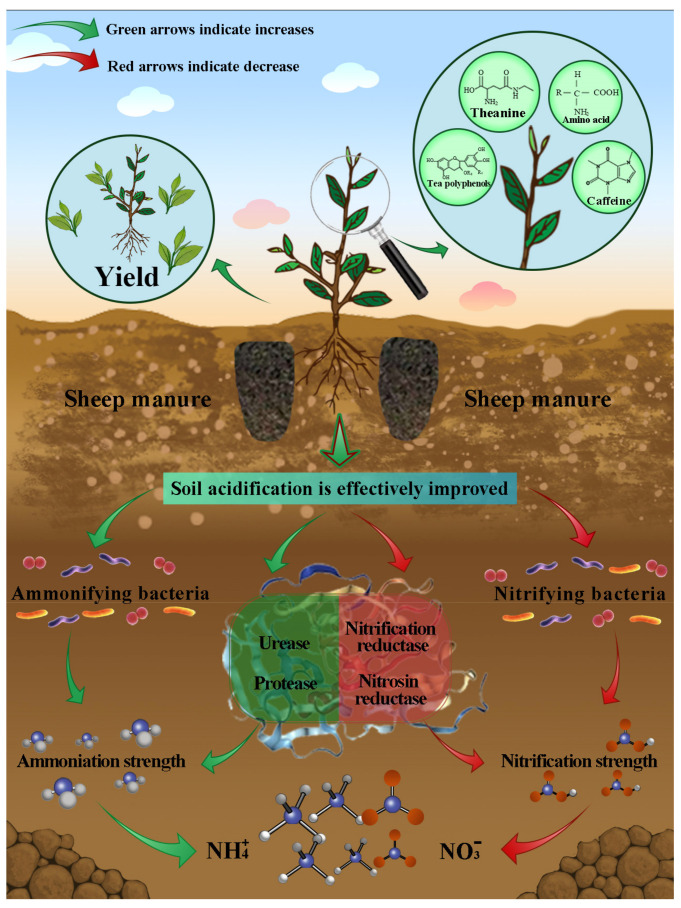
Patterns of effects of sheep manure fertilizer on soil pH, nitrogen transformation, and tea yield and quality in tea plantation.

## Data Availability

The data presented in this study are available as Appendix A.

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
