# Peer review of "Effects of Long-Term Use of Organic Fertilizer with Different Dosages on Soil Improvement, Nitrogen Transformation, Tea Yield and Quality in Acidified Tea Plantations"

_plants, 2022, doi:10.3390/plants12010122_

Round 1

Reviewer 1 Report

The manuscript titled “Effects of long term use of different doses of sheep manure on soil improvement, nitrogen transformation, tea yield and quality in acidified tea plantations” presented interesting information about an important practical basis for the remediation and fertilizer management in acidified tea plantations.

This manuscript is interesting and potentially important topic, and the standard of presenting the research is good. However, there are many notes should be considered before acceptance this manuscript as follows:

-        The title should represent the article's content and facilitate retrieval in indices developed by secondary literature services. A good title (i) briefly identifies the subject, (ii) indicates the purpose of the study, and (iii) gives important and high-impact words early.

-        The abstract must be completely self-explanatory and intelligible in itself. It should include the following: 1. Reason for doing work, including rationale or justification for the research; 2. Objectives and topics covered; 3. Brief description of methods used; 4. Results; 5. Conclusions.

-        M&M: L154-L155: There was no sufficient information about the experimental procedures.In addition, nothing mentioned about the statistical analysis for the data.

-        Discussion: Discussion needs to be rewritten again with providing recent references published in 2021 and 2022.

Author Response

Review 1

The manuscript titled “Effects of long term use of different doses of sheep manure on soil improvement, nitrogen transformation, tea yield and quality in acidified tea plantations” presented interesting information about an important practical basis for the remediation and fertilizer management in acidified tea plantations.

This manuscript is interesting and potentially important topic, and the standard of presenting the research is good. However, there are many notes should be considered before acceptance this manuscript as follows:

-        The title should represent the article's content and facilitate retrieval in indices developed by secondary literature services. A good title (i) briefly identifies the subject, (ii) indicates the purpose of the study, and (iii) gives important and high-impact words early.

Answer: Thanks to the reviewer. The author has revised it.

-        The abstract must be completely self-explanatory and intelligible in itself. It should include the following: 1. Reason for doing work, including rationale or justification for the research; 2. Objectives and topics covered; 3. Brief description of methods used; 4. Results; 5. Conclusions.

Answer: Thanks to the reviewer. The author has revised it.

-        M&M: L154-L155: There was no sufficient information about the experimental procedures.In addition, nothing mentioned about the statistical analysis for the data.

Answer: Thanks to the reviewer. The author has revised it.

-        Discussion: Discussion needs to be rewritten again with providing recent references published in 2021 and 2022.

Answer: Thanks to the reviewer. The author has revised it.

Reviewer 2 Report

Dear colleagues
I read your paper
A very comprehensive set of data is interesting. On the contrary, the shortcoming is that these are not completely new ideas.

However, despite the low novelty, I recommend it for publication.
However, I have a few comments:
I will start unconventionally from the back:
Figure 13 is a relatively redundant general diagram with relatively long-known ideas.
the conclusion corresponds to the paper
I positively assess the discussion on the issue - it is clear and comprehensive

I don't know (according to the rules of the journal) whether it is correct to write the units xx/yy instead of the usual xx.yy-1, if so, I apologize for the redundant information

the jsop data is very clear, orientation in the paper is very easy, the description is clear even for a person who deals with this topic marginally.

your paper is generally a bit more lay than scholarly, but in this case I don't mind. on the other hand, the paper is easy to read

at the beginning of the methodology, please add more data about the location of the experiments, etc.

the introduction is very brief, but contains necessary information, I have no problem with this brevity - on the contrary - many studies are unnecessarily literary

thank you for your work and I wish you much success

Author Response

Review 2

I read your paper
A very comprehensive set of data is interesting. On the contrary, the shortcoming is that these are not completely new ideas.
However, despite the low novelty, I recommend it for publication.

However, I have a few comments:

I will start unconventionally from the back:
Figure 13 is a relatively redundant general diagram with relatively long-known ideas.

Answer: Thanks to the reviewer. Figure 13 is a summary of the results obtained in this study to facilitate the reader to better understand the study.

the conclusion corresponds to the paper

Answer: Thanks to the reviewer.

I positively assess the discussion on the issue - it is clear and comprehensive

Answer: Thanks to the reviewer.

I don't know (according to the rules of the journal) whether it is correct to write the units xx/yy instead of the usual xx.yy-1, if so, I apologize for the redundant information

Answer: Thanks to the reviewer. The authors will consult with the journal editor on whether this writing style needs to be revised.

the jsop data is very clear, orientation in the paper is very easy, the description is clear even for a person who deals with this topic marginally.

Answer: Thanks to the reviewer.

your paper is generally a bit more lay than scholarly, but in this case I don't mind. on the other hand, the paper is easy to read at the beginning of the methodology, please add more data about the location of the experiments, etc.

Answer: Thanks to the reviewer, the author has added.

the introduction is very brief, but contains necessary information, I have no problem with this brevity - on the contrary - many studies are unnecessarily literary

Answer: Thanks to the reviewer.

Reviewer 3 Report

·     Well-designed research, appropriately written, minor corrections needed!

1. The abstract is too long (more than 350 words) it should be up to 200 words long with only one paragraph.  The author followed the style of a structured abstract, but I think you don't have to show all the results just mention the most important results.

2. Grammar check is needed and some words have typo mistakes. For exp. L409 and L422.

3. L48 and 51 it is better to remove the names and make the sentence in the passive tense, and put the reference at the end of the sentence. As well, for L430, L437, L448, and L482 please put the name of the method not the refer. and put the reference at the end of the sentence.

4. L54-57 rewrite this sentence with different synonyms.

5.  L77-81 rewrite this sentence it’s not clear.

6. Could you provide more references?

Author Response

Review 3

Well-designed research, appropriately written, minor corrections needed!

  1. The abstract is too long (more than 350 words) it should be up to 200 words long with only one paragraph.  The author followed the style of a structured abstract, but I think you don't have to show all the results just mention the most important results.

Answer: Thanks to the reviewer. The author has made the appropriate changes

  1. Grammar check is needed and some words have typo mistakes. For exp. L409 and L422.

Answer: Thanks to the reviewer. The author has revised it.

  1. L48 and 51 it is better to remove the names and make the sentence in the passive tense, and put the reference at the end of the sentence. As well, for L430, L437, L448, and L482 please put the name of the method not the refer. and put the reference at the end of the sentence.

Answer: Thanks to the reviewer. The author has revised it.

  1. L54-57 rewrite this sentence with different synonyms.

Answer: Thanks to the reviewer. The author has revised it.

  1. L77-81 rewrite this sentence it’s not clear.

Answer: Thanks to the reviewer. The author has revised it.

  1. Could you provide more references?

Answer: Thanks to the reviewer. The author has revised it.

Round 2

Reviewer 2 Report

I agree

Author Response

Thanks to the reviewers